# CRISPR/Cas9 Knock-Out in Primary Neonatal and Adult Cardiomyocytes Reveals Distinct cAMP Dynamics Regulation by Various PDE2A and PDE3A Isoforms

**DOI:** 10.3390/cells12111543

**Published:** 2023-06-04

**Authors:** Egor B. Skryabin, Kirstie A. De Jong, Hariharan Subramanian, Nadja I. Bork, Alexander Froese, Boris V. Skryabin, Viacheslav O. Nikolaev

**Affiliations:** 1Institute of Experimental Cardiovascular Research, University Medical Center Hamburg-Eppendorf, 20246 Hamburg, Germany; egor.skryabin@stud.uke.uni-hamburg.de (E.B.S.); k.de-jong@uke.de (K.A.D.J.); h.subramanian@uke.de (H.S.); n.bork@uke.de (N.I.B.); a.froese@uke.de (A.F.); 2German Center for Cardiovascular Research (DZHK), Partner Site Hamburg/Kiel/Lübeck, 20246 Hamburg, Germany; 3Core Facility Transgenic Animal and Genetic Engineering Models (TRAM), University of Münster, 48149 Münster, Germany; skryabi@uni-muenster.de

**Keywords:** cAMP, phosphodiesterases, cardiomyocyte, CRISPR/Cas9, isoform

## Abstract

Cyclic nucleotide phosphodiesterases 2A (PDE2A) and PDE3A play an important role in the regulation of cyclic adenosine monophosphate (cAMP) and cyclic guanosine monophosphate (cGMP)-to-cAMP crosstalk. Each of these PDEs has up to three distinct isoforms. However, their specific contributions to cAMP dynamics are difficult to explore because it has been challenging to generate isoform-specific knock-out mice or cells using conventional methods. Here, we studied whether the CRISPR/Cas9 approach for precise genome editing can be used to knock out *Pde2a* and *Pde3a* genes and their distinct isoforms using adenoviral gene transfer in neonatal and adult rat cardiomyocytes. Cas9 and several specific gRNA constructs were cloned and introduced into adenoviral vectors. Primary adult and neonatal rat ventricular cardiomyocytes were transduced with different amounts of Cas9 adenovirus in combination with PDE2A or PDE3A gRNA constructs and cultured for up to 6 (adult) or 14 (neonatal) days to analyze PDE expression and live cell cAMP dynamics. A decline in mRNA expression for PDE2A (~80%) and PDE3A (~45%) was detected as soon as 3 days post transduction, with both PDEs being reduced at the protein level by >50–60% in neonatal cardiomyocytes (after 14 days) and >95% in adult cardiomyocytes (after 6 days). This correlated with the abrogated effects of selective PDE inhibitors in the live cell imaging experiments based on using cAMP biosensor measurements. Reverse transcription PCR analysis revealed that only the PDE2A2 isoform was expressed in neonatal myocytes, while adult cardiomyocytes expressed all three PDE2A isoforms (A1, A2, and A3) which contributed to the regulation of cAMP dynamics as detected by live cell imaging. In conclusion, CRISPR/Cas9 is an effective tool for the in vitro knock-out of PDEs and their specific isoforms in primary somatic cells. This novel approach suggests distinct regulation of live cell cAMP dynamics by various PDE2A and PDE3A isoforms in neonatal vs. adult cardiomyocytes.

## 1. Introduction

Cyclic nucleotide phosphodiesterases (PDEs) form a large superfamily of enzymes that hydrolyze and hence play an important role in the control of intracellular cyclic adenosine monophosphate (cAMP) and cyclic guanosine monophosphate (cGMP) levels [1,2,3]. By restricting their diffusion, these enzymes are one of the key factors orchestrating the compartmentalization of cyclic nucleotides in the so-called subcellular micro- or nanodomains [4,5,6].

Out of 11 known PDE families with >100 individual isoforms, the PDE2 and PDE3 families are unique in their ability for cGMP-dependent regulation of cAMP levels via the so-called cGMP/cAMP cross-talk, with the PDE2A and PDE3A subfamilies predominantly expressed in mammalian heart muscle cells, also called cardiomyocytes [7,8]. cGMP can bind to an N-terminal allosteric domain of PDE2A which leads to increased enzyme activity enabling the so-called negative cGMP/cAMP cross-talk. The catalytic domain of PDE3A has a high affinity for cGMP which acts as a competitive substrate inhibiting cAMP degradation and enabling a positive cGMP/cAMP cross-talk.

Each of these PDEs has up to three distinct isoforms generated by alternative splicing of relatively short N-terminal exons [1,8]. However, their specific contributions to cAMP dynamics are difficult to explore because of two reasons. First, chemical inhibitors can barely distinguish between the isoforms. Second, due to the short alternative exon structure at the N-terminus of these PDEs, it has been difficult to generate isoform-specific knock-out mice or cells using conventional methods. Therefore, it is rather challenging to study how individual PDE isoforms regulate cyclic nucleotide dynamics and compartmentalization at the subcellular level.

Here, we developed a CRISPR/Cas9-based genome editing approach for efficient functional knock-out of multiple and individual PDE2A and PDE3A isoforms in primary rat cardiomyocytes. This technique should be useful for many studies focusing on the contribution of individual PDE isoforms to cardiac physiology and pathophysiology.

## 2. Materials and Methods

### 2.1. Chemicals and Kits

The Gateway LR Clonase Enzyme Kit was from Thermo Fischer Scientific (Dreieich, Germany). The cilostamide was from Santa Cruz Biotechnology (Santa Cruz, CA, USA). BAY 60-7550, and all other chemicals were from Sigma-Aldrich (Deisenhofen, Germany).

### 2.2. Cloning of CRISPR/Cas9 Constructs and Adenovirus Generation

The pDNR221_U6gRNA vector for individual gRNAs was assembled from two different elements: (1) the EcoRI-SalI DNA fragment, containing the U6 promoter, the lacZ gene, and the gRNA (781 bp) were isolated from the pU6gRNA plasmid and ligated together with (2) the EcoRI-HindIII DNA fragment, containing the cytomegalovirus (CMV) promoter and mMaroon fluorescent protein gene, into the pDONR221 vector backbone (Thermo Fisher Scientific, Dreieich, Germany) via the XhoI and HindIII restriction sites. The final plasmid map and sequence information are presented in Appendix A.

The pDNR221 vector encoding Cas9 (pDNR221_Cas9) was cloned based on the same pDONR221 vector backbone to contain the following three additional elements: (1) the PCR fragment encoding the ×3 STOP signals cut with HindIII + MluI (520 bp), (2) the 4.8 kb DNA fragment MluI—AgeI containing the Cas9 gene from plasmid pCas9_DAPCEL, and (3) the PCR fragment (403 bp) AgeI—XhoI containing the polyA signal. The final plasmid map and sequence information are presented in Appendix A.

Newly generated pDNR221 vectors encoding Cas9 or individually cloned gRNA sequences were recombined into the adenoviral vector pAd using gateway system cloning (LR reaction, according to the manufacturer’s protocol) and transfected into 293A cells (Thermo Fisher Scientific) using lipofection. The adenoviruses were amplified and purified over the CsCl gradient as previously described [9].

### 2.3. Cardiomyocyte Isolation and Culture

Neonatal cardiomyocytes were isolated from postnatal day 1–2 rat pups as previously described [10] and plated on gelatin-coated plates (for immunoblots) or glass cover slides (for live cell imaging). One day after isolation, the culture medium was changed, and adenoviruses were added at the indicated multiplicity of infection (MOI). Adult rat ventricular cardiomyocytes were isolated, plated on laminin-coated plates or cover slides, transduced with adenoviral vectors, and cultured as previously described [11].

### 2.4. Immunoblot Analysis

Cells were scrapped off the plates and homogenized in a buffer containing: 10 mM HEPES, 300 mM sucrose, 150 mM NaCl, 1 mM EGTA, 2 mM CaCl_2,_ and 1% Triton-X. The proteins were quantified using Pierce BCA protein assay (Thermo Fisher Scientific). The samples were denatured at 70 °C for 10 min, and 10–20 μg of total protein per lane was subjected to 10% SDS-PAGE and immunoblot analysis using an anti-PDE2A antibody (Fabgennix, Frisco, TX, USA), a custom-made rabbit polyclonal PDE3A antibody (kindly provided by Chen Yan, University of Rochester, Rochester, NY, USA), a Cas9 antibody (Cell Signaling Technology), and a mouse monoclonal anti-Glyceraldehyde-3-Phosphate Dehydrogenase (GAPDH) antibody (HyTest, Turku, Finland). All blots were scanned and analyzed by ImageJ 1.44 software for uncalibrated optical density.

### 2.5. Live Cell Imaging

For Förster resonance energy transfer (FRET) measurements, cover slides with adherent cardiomyocytes were mounted in an Attofluor microscopy chamber and maintained in FRET buffer containing 144 mM NaCl, 5.4 mM KCl, 1 mM MgCl_2_, 1 mM CaCl_2_, and 10 mM HEPES at pH 7.3. Live cell imaging was performed using a custom-made FRET microscopy system built around a Leica DMI 3000B microscope (Leica Microsystems, Wetzlar, Germany) equipped with a 63×/1.40 oil-immersion objective [12]. The donor fluorophore-enhanced cyan fluorescent protein (CFP) was excited with 440 nm light at a 5–10 s interval using a CoolLED single-wavelength light emitting diode. Emitted light was separated into CFP and enhanced yellow fluorescent protein (YFP) channels using a DV2 DualView and detected using an OptiMOS charge-coupled device camera (both from Photometrics, Surrey, BC, Canada). The FRET system was controlled for data acquisition using Icy 2.4.0 (Institut Pasteur and France-BioImaging, Paris, France) and MicroManager 1.4.3 (Laboratory for Optical and Computational Instrumentation at the University of Wisconsin, Madison, WI, USA) software as previously described [13]. After reaching a stable baseline, the cardiomyocytes were treated with different compounds diluted in the FRET buffer to stimulate cellular cAMP responses. Offline data analysis was performed using ImageJ, Microsoft Excel 2016 (Microsoft, Munich, Germany), and GraphPad Prism 7.0 (GraphPad Software, Boston, MA, USA) software.

### 2.6. Statistics

Normal distribution was tested by the Kolmogorov–Smirnov test, and differences between the groups were analyzed using one-way ANOVA followed by the Sidak test for multiple-group comparisons or the Mann–Whitney test, as appropriate. FRET imaging data obtained using multiple cells from several animals/isolations were analyzed using a mixed ANOVA followed by Wald’s Chi-squared test. The statistical analysis was performed using GraphPad Prism 7.0 software.

## 3. Results

We sought to employ the CRISPR/Cas9 system to knock out all or individual PDE2A and PDE3A isoforms in the cardiomyocytes. Therefore, we first designed pairs of gRNAs targeting either exons encoding the catalytic domains of these PDEs which are similar in all isoforms and should therefore generate “complete” knock-outs for all PDE2A (PDE2A_8_3 and PDE2A_8_1) or PDE3A (PDE3A_12_1 and PDE3A_12_2) isoforms. Next, we designed gRNA pairs for exons used for alternative splicing of PDE2A (Figure 1A) and PDE3A (Figure 1B) to enable specific 2A1, 2A2, 2A3, 3A1, and 3A2 isoform knock-outs. All gRNAs were individually cloned under the control of the U6 promoter into the pDNR221_U6gRNA vector also expressing a red fluorescent protein mMaroon and further recombined into an adenovirus type 5 vector (Appendix A) to generate high titer adenoviruses.

To test these new viral vectors, we isolated neonatal rat cardiomyocytes and transduced them with Cas9 (the titer was preselected based on expression analysis in these cells transduced with different amounts of Cas9 virus—Appendix A) and various amounts of gRNAs for total PDE2A or PDE3A knock-outs. Sequencing of the PDE2A/3A encoding genes proved the expected genomic rearrangements including specific deletion for a part of exon 8 in PDE2A and a part of exon 12 of the PDE3A sequences (Appendix A). Approximately 14 days after transduction, we lysed the cells and performed an immunoblot analysis of PDE expression. We could detect a >50% reduction in PDE2A and PDE3A protein content at the most optimal gRNA MOI 3 and 10, respectively (Figure 2). Although virtually all cells were fluorescent in the mMaroon channel (Appendix A), suggesting the expression of gRNAs, the observed incomplete knock-out could be the result of variable/submaximal efficiencies for some gRNAs and/or extensive proliferation of some cells after transduction. Moreover, we transduced ventricular neonatal rat cardiomyocytes with Cas9 together with different amounts of gRNA adenoviruses (MOI 0.5; 3; 10) to measure mRNA expression levels 3 days after transduction using quantitative reverse transcription (RT)-qPCR. We detected an 81 ± 2% (mean ± s.e.m., n = 3) reduction in PDE2A and 43 ± 8% (mean ± s.e.m., n = 3) reduction in PDE3A mRNA expression, suggesting that already after 3 days, there is a strong reduction in mRNA levels, with PDE3A mRNA declining more slowly, potentially due to its longer half-life.

To determine the functional effects of these knock-outs, we measured the elicited cAMP responses to beta-adrenergic receptor stimulation and PDE inhibition in these cardiomyocytes which were additionally transduced with an adenovirus encoding for the cytosolic cAMP biosensor Epac1-camps [11]. Upon cAMP binding, this sensor shows a rapid decrease in the YFP/CFP acceptor/donor fluorescence ratio which is indicative of rising intracellular cAMP concentrations. In this work, virtually all measured cardiomyocytes co-expressing the cAMP biosensor together with gRNAs and Cas9 exhibited abrogated responses to the selective PDE2 or PDE3 inhibitors BAY 60-7550 (Figure 3A–D) and Cilostamide (Figure 3E–H), respectively, suggesting an expected effective deletion of all PDE2A or PDE3A isoforms at the functional level. Transduction with Cas9 alone led to somewhat lower BAY and Cilo responses as compared to NT controls, which is theoretically possible due to the adenoviral treatment; however, these differences did not reach statistical significance (Figure 3D,H).

To assess the efficiency of the system in terminally differentiated adult rat cardiomyocytes, we transduced these cells with various amounts of gRNA vectors together with Cas9 adenovirus used at MOI 300 (as an optimal pre-tested concentration, see Appendix A). Approximately 6 days after transduction, the cells lost the vast majority of the PDE3A or PDE2A protein, as confirmed by the immunoblot analysis (Figure 4).

Functionally, this correlated with completely abolished responses to the PDE2 but not the PDE3 inhibitor in the PDE2A knock-out cells (Figure 5A–C, Appendix A) and abrogated Cilostamide but not the BAY 60-7550 response in the PDE3A knock-out cardiomyocytes (Figure 5D–F, Appendix A). Interestingly, PDE2A knock-out also led to an increased ISO response, suggesting tight regulation of beta-adrenergic receptor/cAMP responses by this PDE (Figure 5C).

Various PDE2A and PDE3A isoforms have been reported to be expressed in cardiomyocytes. Interestingly, when performing RT-PCR with isoform-specific primers for cDNA made from neonatal and adult rat cardiomyocyte RNA, only the PDE2A2 isoform was detected in the neonatal cardiomyocytes, while all three PDE2A isoforms were detected in the adult cardiomyocytes (Figure 6).

To test the possibility of selective isoform knock-out, we co-infected adult rat cardiomyocytes with Cas9 and gRNA adenoviruses designed to knock out the best characterized PDE2A2 (gRNAs PDE2A_21_1 and PDE2A_21_2, see Figure 1) and PDE2A3 (gRNAs PDE2A_31_1 and PDE2A_31_3) isoforms, either individually or both together, and measured FRET responses to BAY 60-7550 at day 6 after transduction. Compared to the control cells expressing Cas9 only, the cells with single PDE2A2 or PDE2A3 knock-outs showed only a marginal if any decrease in PDE2 inhibitor response, suggesting that multiple isoforms contribute to total PDE2A activity in adult cardiomyocytes. Interestingly, combined knock-out of both PDE2A2 and PDE2A3 led to almost completely abrogated BAY 60-7550 response (Figure 7), confirming that both isoforms form the vast majority of the active PDE2A pool regulating beta-adrenoceptor/cAMP responses in these cells.

## 4. Discussion

In this study, we employed a CRISPR/Cas9 system to knock out multiple or individual PDE2A and PDE3A isoforms in neonatal and adult rat ventricular cardiomyocytes. These PDEs together with other PDE families play central roles in cAMP compartmentalization for which cardiomyocytes have provided an important paradigm [2,6]. In addition, these PDEs are central for the cGMP/cAMP cross-talk which is known to regulate cardiac function and disease [7,8,14]. Despite the importance of both PDE families, the role of individual isoforms has been difficult to study because chemical PDE inhibitors cannot fully differentiate between them, with inhibitors having only a relatively small difference in affinity to individual isoforms of the same PDE subfamily [15]. One elegant approach to overcome this problem has been developed based on the overexpression of catalytically inactive PDE2A and PDE3A isoforms [2,6] which act in a dominant negative fashion by displacing endogenous PDEs from their subcellular locations [16].

Here, we generated gRNA adenoviruses to knock out either all PDE2A or PDE3A isoforms by targeting their common catalytic domain sequences or to target unique N-terminal domain sequences generated in each individual isoform by alternative splicing. We were able to show that the high transduction efficiency of these viruses leads to a rapid decay in mRNA expression after 3 days, expected CRISPR-induced genomic rearrangements in the targeted locus, and loss of most PDE protein after 6 days in adult cardiomyocytes and 14 days in neonatal cardiomyocytes post transduction. Despite only partial downregulation of protein expression in neonatal cells, which might be due to variable gRNA efficiency, Cas9 expression, or overgrowth of non-transduced cells, at the functional level, virtually all cells showed completely abolished effects of PDE2 and PDE3 inhibitors on intracellular cAMP levels (see Figure 3). In adult cardiomyocytes, the decline in PDE protein levels already at 6 days was somewhat greater, likewise with abrogated cAMP responses to PDE inhibitor application (see Figure 4 and Figure 5). Live cell imaging experiments can be more reliable in this regard because they can be performed after the control of red fluorescence (originating from mMaroon which labels gRNA-expressing cells) in each measured cell.

In terms of individual PDE2A isoform knock-outs, we could confirm at the functional level using live cell cAMP imaging that both PDE2A2, which is a well characterized mitochondrial isoform [17], and PDE2A3, known as a plasma membrane targeted isoform which is important for cardiac function and arrhythmia protection [18,19], act in concert to regulate cAMP levels in adult rat cardiomyocytes (see Figure 7). Our data also suggest potentially different regulation of cAMP by various PDE2A and PDE3A isoforms in neonatal vs. adult cardiomyocytes. While neonatal cells rely only on the mitochondrial PDE2A2 isoform, PDE2-dependent regulation of cAMP in adult myocytes is more complex and involves all three isoforms localized at different subcellular compartments which has to be more specifically addressed in future studies.

Our approach can be potentially applied to various cell types, especially primary cell types which can be effectively transduced with adenoviral vectors to study the role of individual isoforms in all possible PDE subfamilies. For example, in the case of cardiomyocytes, apart from PDE2A and PDE3A, the role of numerous PDE4 subfamily isoforms can be addressed. This PDE family comprises at least two dozen different splice variants which regulate local cAMP functions in multiple subcellular microdomains [1,3]. Not only can cAMP and cGMP dynamics be measured by live cell imaging in KO cells, our system makes it possible to assess functional responses through multiple approaches such as immunoblotting for downstream kinase substrate phosphorylation, contractility measurements, regulation of arrhythmogenic events, pathological cell growth, and cell survival.

As a limitation, our study could not verify or quantify the efficiency of individual PDE2A isoform knock-outs at the protein level because isoform-specific antibodies are not available for all three isoforms, and they also cannot be distinguished with our pan-PDE2A antibody due to their similar size. In addition, we could not assess cell-to-cell variability of the knock-out due to the lack or accessibility of suitable techniques.

## 5. Conclusions

In conclusion, we have established and characterized a versatile, easily accessible system to generate knock-outs of single PDE2A and PDE3A isoforms in somatic cells which can be applied for further functional studies of individual PDE isoform contributions to cardiomyocytes function and potentially also pathophysiological responses using in vitro/ex vivo settings. Furthermore, this approach can be potentially extended to other PDE families and other cell types.

## Figures and Tables

**Figure 1 cells-12-01543-f001:**
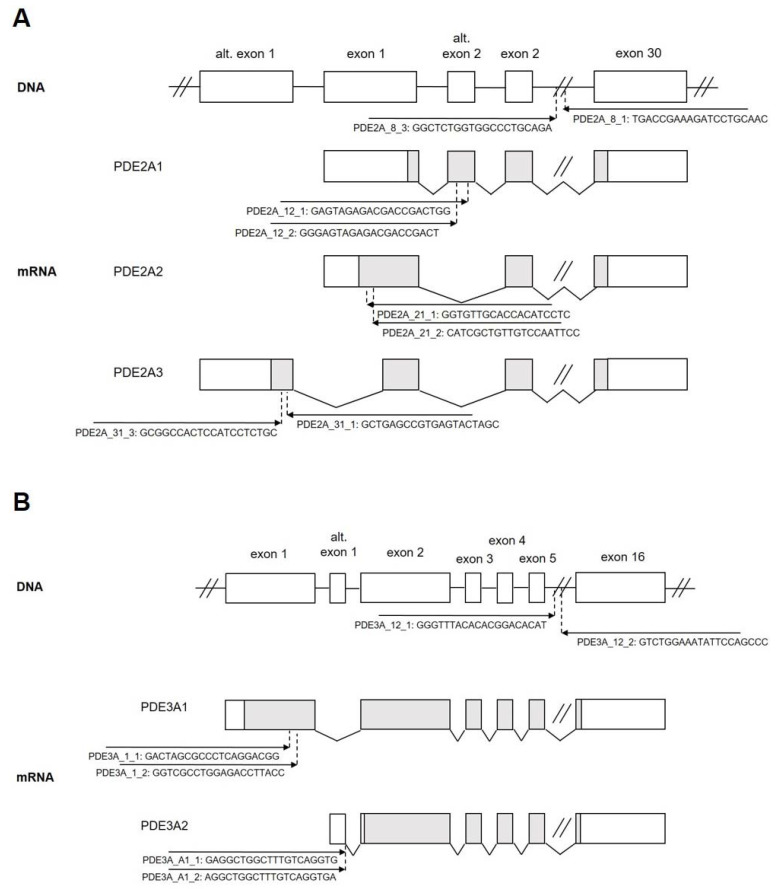
Genomic structure of rat (**A**) PDE2A and (**B**) PDE3A genes. Alternative splicing giving rise to individual isoforms is shown. The positions, orientation, and sequences of the gRNA constructs used in this study are indicated with arrows.

**Figure 2 cells-12-01543-f002:**
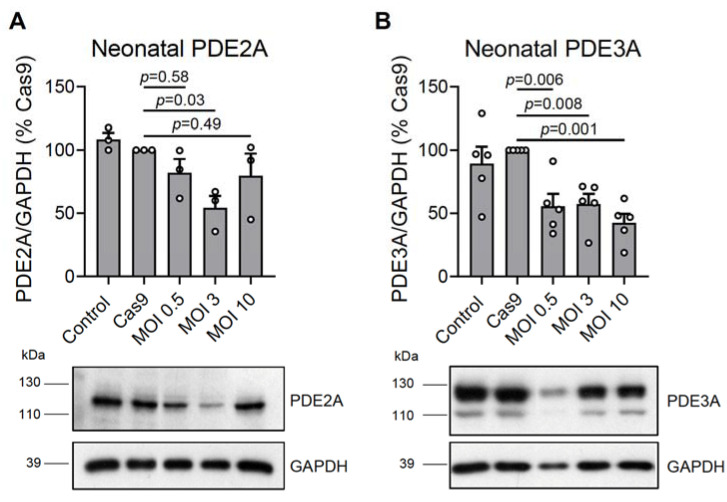
Immunoblot analysis of (**A**) PDE2A and (**B**) PDE3A expression in rat neonatal cardiomyocytes. The cultured cells were not transduced (NT, control) or were transduced with Cas9 (MOI 1) or with Cas9 and gRNA constructs targeting all PDE2A (PDE2A_8_3 and PDE2A_8_1) or PDE3A (PDE3A_12_1 and PDE3A_12_2) isoforms at various MOIs. The PDE expression data were normalized to GAPDH and shown as % of Cas9 samples. Representative blots and their analysis are shown. The PDE3A signal in the MOI 0.5 sample appears fainter than for other samples because of lower protein loading, as evidenced by the GAPDH signal. *p*-values were calculated using one-way ANOVA followed by the Sidak multiple comparisons test.

**Figure 3 cells-12-01543-f003:**
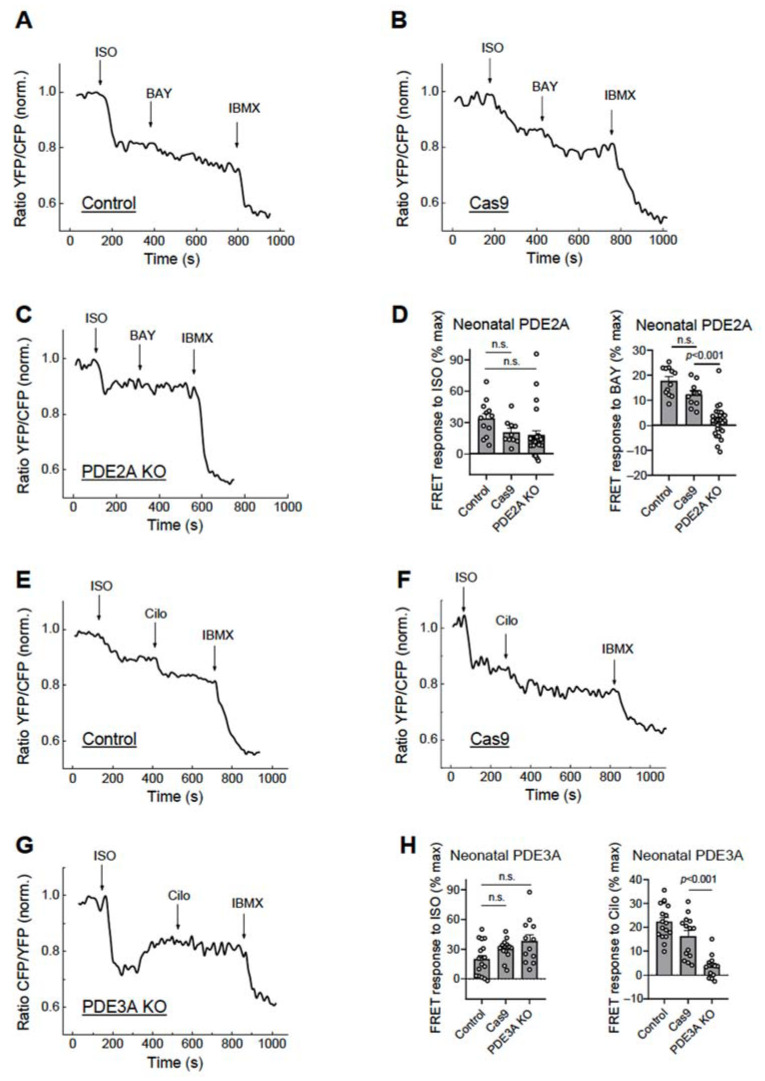
Live cell imaging analysis of PDE2 and PDE3 inhibitor effects in neonatal cardiomyocytes. The neonatal rat ventricular cardiomyocytes were either not transduced (NT, control) or transduced for 14 days with Cas9 alone (Cas9) or Cas9 plus gRNA adenoviruses as described in Figure 2. The effect of the PDE2 inhibitor BAY 60-7550 (100 nM) in PDE2A-KO cells (**A**–**D**) and the effect of the PDE3 inhibitor Cilostamide (Cilo, 10 µM) in the PDE3A-KO cells (**E**–**H**) were measured after beta-adrenergic stimulation with 10 nM isoproterenol (ISO), followed by the maximal response reached with 100 µM of the pan-PDE inhibitor 3-Isobutyl-1-methylxanthine (IBMX). The time points of drug application are indicated by arrows. The drugs were applied after reaching a stable baseline for each treatment. Representative FRET traces and data analysis for ISO, BAY, and Cilo effects (expressed as % of maximal response). *p*-values were calculated using a mixed ANOVA followed by a Chi-squared test; n.s.—not significant.

**Figure 4 cells-12-01543-f004:**
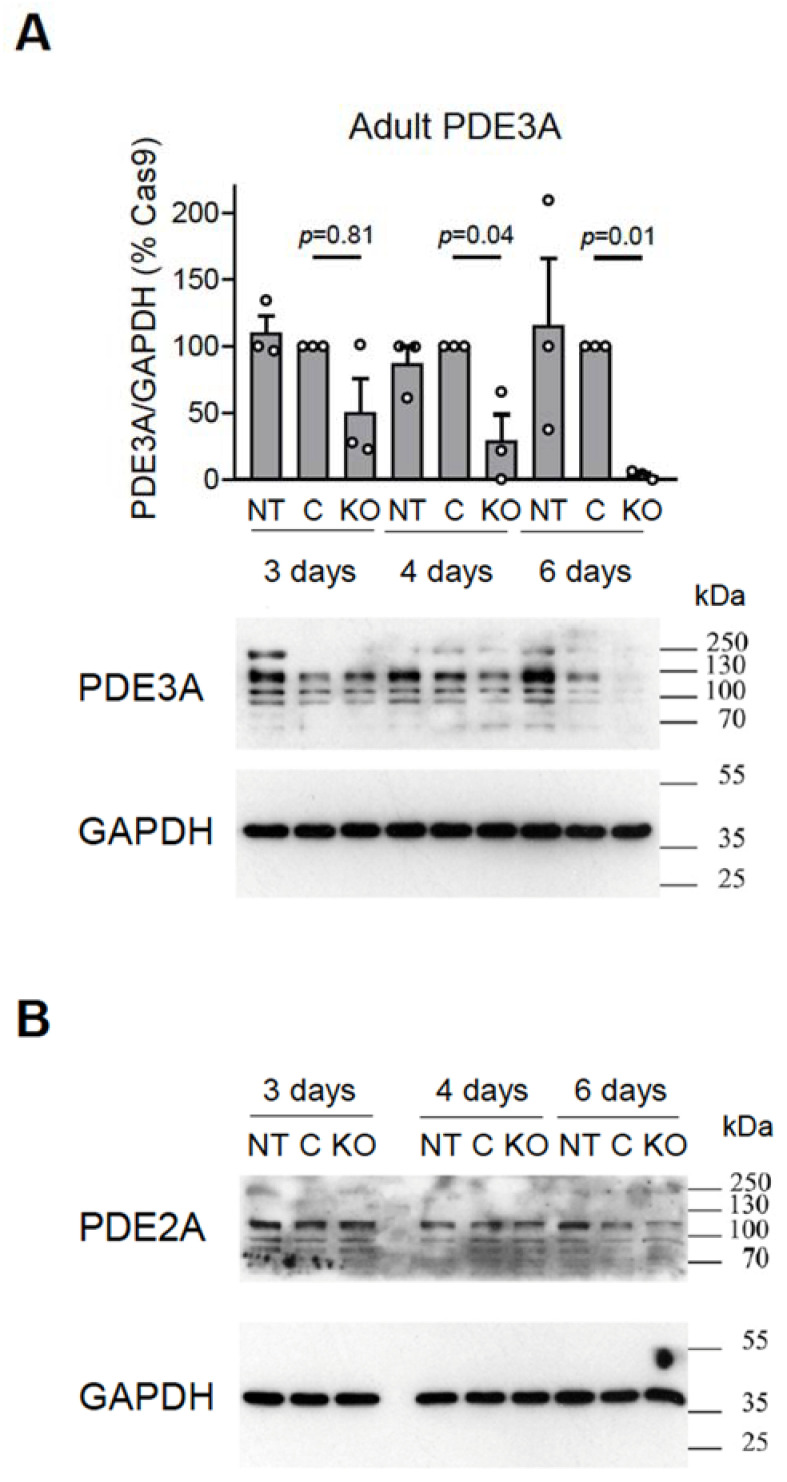
Immunoblot analysis of (**A**) PDE3A and (**B**) PDE2A expression in adult rat cardiomyocytes. The cultured cardiomyocytes were not transduced (NT) or were transduced with Cas9 (MOI 300) without (C) or with gRNA constructs (KO) targeting all PDE2A ((**A**): PDE3A_12_1 and PDE3A_12_2) or PDE3A ((**B**): PDE2A_8_1 and PDE2A_8_3) isoforms at MOI 30 and harvested at days 3, 4, and 6 after transduction for immunoblot analysis. Representative blots and analysis of PDE3A expression as compared to the Cas9 transduced control (C) are shown. The PDE expression data were normalized to GAPDH and are shown as % of the Cas9 control samples. *p*-values were calculated using one-way ANOVA followed by the Sidak multiple comparison test.

**Figure 5 cells-12-01543-f005:**
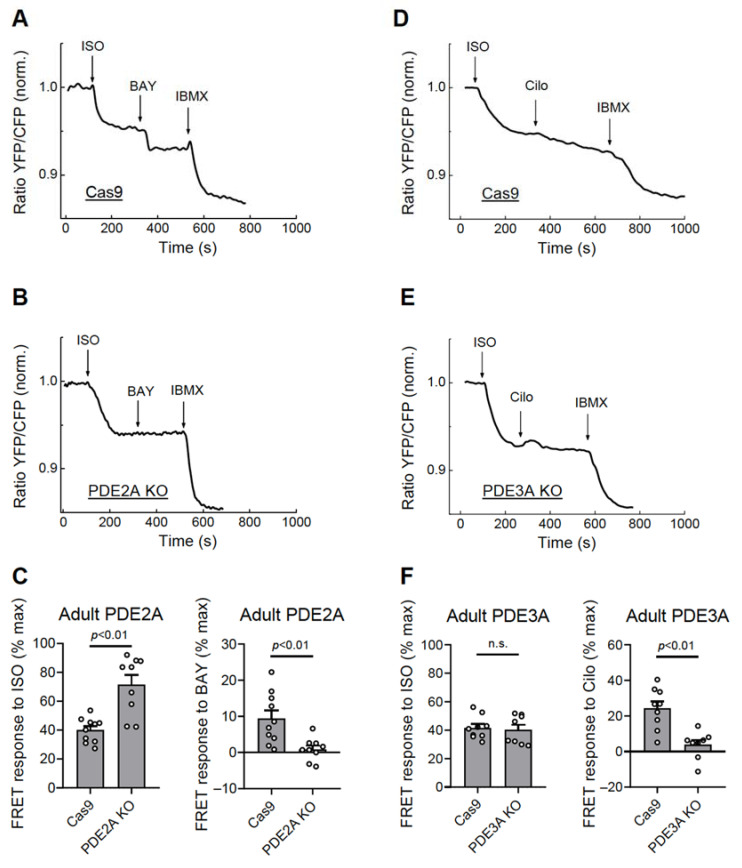
Live cell imaging analysis of PDE2 and PDE3 inhibitor effects in adult cardiomyocytes. The adult rat ventricular cardiomyocytes were transduced for 6 days with Cas9 or Cas9 plus gRNA adenoviruses as described in Figure 4. The effect of the PDE2 inhibitor BAY 60-7550 (BAY, 100 nM) in the PDE2A-KO cells (**A**–**C**) and the effect of the PDE3 inhibitor Cilostamide (Cilo, 10 µM) in the PDE3A-KO cells (**D**–**F**) were measured after beta-adrenergic stimulation with 100 nM ISO as described in Figure 3. Representative FRET traces and data analysis for ISO, BAY, and Cilo responses are shown. The *p*-values were from a mixed ANOVA followed by a Chi-squared test; n.s.—not significant.

**Figure 6 cells-12-01543-f006:**
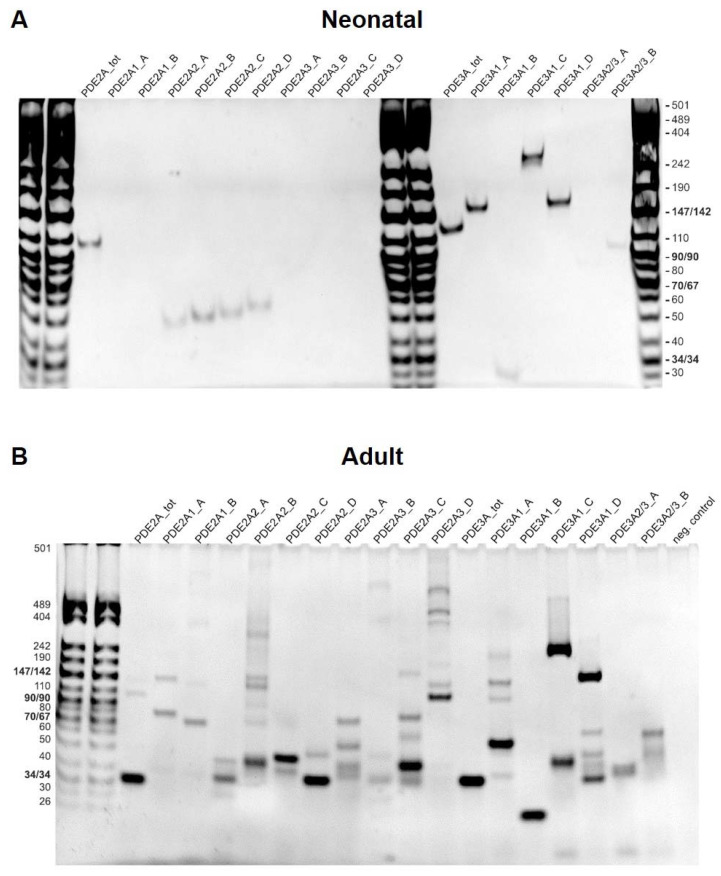
PDE2A and PDE3A isoform expression in neonatal and adult cardiomyocytes. RT-PCR analysis was performed for neonatal (**A**) and adult (**B**) rat ventricular cardiomyocytes. Several primer combinations as described in Appendix A were designed to detect total PDE2A and PDE3A mRNA or specific PDE2A and PDE3A isoform expression. PCR products were loaded on a 6% polyacrylamide gel. To detect all isoforms even at low expression, we performed 14 cycles of touchdown PCR followed by 35 cycles of regular PCR for both neonatal and adult cells. This may explain the high number of potentially unspecific bands appearing next to the most prominent specific band (see Appendix A for expected size), especially in the case of adult cardiomyocytes where the expression of RNAs prone to unspecific amplification might be higher.

**Figure 7 cells-12-01543-f007:**
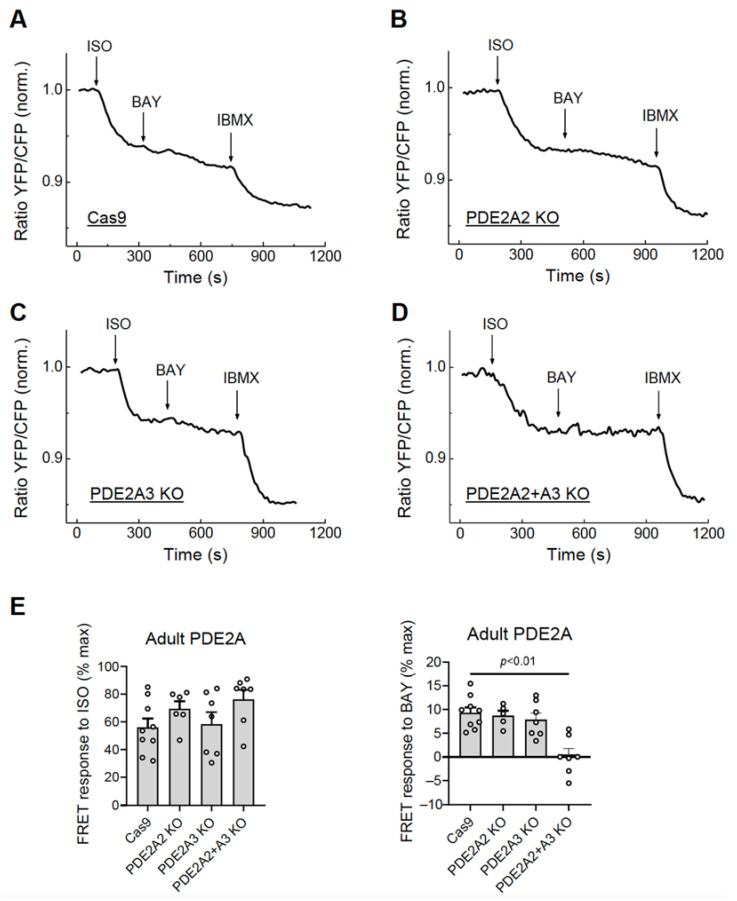
Live cell imaging analysis of PDE2 inhibitor effects in adult cardiomyocytes with knock-outs of individual PDE2A isoforms. The adult rat ventricular cardiomyocytes were transduced for 6 days with Cas9 or Cas9 plus gRNA adenoviruses for either PDE2A2, PDE2A3, or both isoforms together. The effect of the PDE2 inhibitor BAY 60-7550 (BAY, 100 nM) in PDE2A-KO cells were measured after beta-adrenergic stimulation with 100 nM ISO as described in Figure 5. Representative FRET traces (**A**–**D**) and data analysis of BAY and ISO responses (**E**) are shown. The *p*-value was calculated by a mixed ANOVA followed by a Chi-squared test; all other differences in E were not significant.

## Data Availability

Data are contained within the article or Appendix A. Raw data and materials are available from the authors upon reasonable request.

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
