# Peer review of "CRISPR/Cas9 Knock-Out in Primary Neonatal and Adult Cardiomyocytes Reveals Distinct cAMP Dynamics Regulation by Various PDE2A and PDE3A Isoforms"

_cells, 2023, doi:10.3390/cells12111543_

Round 1

Reviewer 1 Report

The manuscript from Skyrabin and colleagues describes the setup and functional test of gene editing based downregulation of specific PDE isoforms, namely PDE2A and PDE3A, in cultured/freshly isolated primary cardiomyocytes.

The experimental design is clear and well laid out in the manuscript, and methods are sufficiently described.

The scientific approach uses the quite standardised approach of real-time fluorescence study of cAMP variations in cells, which exploits GFP sensors extensively validated by the authors and other laboratories. 

The functional pharmacologic experiments are neat and not difficult to understand, making the text readable and easy to follow even for the non-expert colleague. Results are, likewise, well described and in vitro fluorescence studies provide a good support to the molecular/biochemical effects shown in the first two figures.

While data are not surprising but rather confirmatory, the value of the work is clear as it shows a method to overcome pharmacology in the functional study of specific PDE isoforms. 

one concern of this reviewer regards the FRET experiments: 

1- in Figure 1, response to PDE2 inhibition in cells expressing Cas9 seems decreased, or frankly abolished, which I wouldn't expect, as reported in the summary graph in 1D. 

While I do believe quantification made for Figure 1D, it seems the curve shown in 1B is not representative of the values indicated in the figure, OR that cAMP changes upon pharmacologic stimulation have not been quantified properly. Authors should double check image processing and analysis.

2- Upon downregulation of PDE2 and 3 (or either one), I would expect effects on the amplitude of cAMP responses  to ISO. Once again, "representative" panels do not help to understand this. Have the authors quantified ISO responses in PDE KD cells? Even cAMP dynamics could be different, but authors seem not to have cared for this. They could increase cAMP imaging data analysis.

3- the same concept applies to IBMX responses, which are expected to reflect downregulation of specific PDE isoforms - and would therefore be different in cells already devoid of either PDE2 or PDE3

4- can the authors comment as to why PDE2 or PDE3 downregulation does not impact on basal cAMP concentration? 

Author Response

We thank the Reviewer a lot for his/her very positive evaluation of our manuscript. We did our best to address all concerns as detailed point-by-point below.

one concern of this reviewer regards the FRET experiments: 

1- in Figure 3, response to PDE2 inhibition in cells expressing Cas9 seems decreased, or frankly abolished, which I wouldn't expect, as reported in the summary graph in 3D. 

While I do believe quantification made for Figure 1D, it seems the curve shown in 1B is not representative of the values indicated in the figure, OR that cAMP changes upon pharmacologic stimulation have not been quantified properly. Authors should double check image processing and analysis.

We thank the Reviewer for raising this important point. The trace originally shown in Figure 3B was not very nice, also the traces in Fig. 3A, B and C were not very representative in terms of the ISO response amplitude. Therefore, we have found better representative traces and now added them to Fig 3A,B and C. Cas9 expression itself did not cause significant reduction of the BAY response which was seen only in Cas9+gRNA transduced (PDE2-KO) cells.

2- Upon downregulation of PDE2 and 3 (or either one), I would expect effects on the amplitude of cAMP responses  to ISO. Once again, "representative" panels do not help to understand this. Have the authors quantified ISO responses in PDE KD cells? Even cAMP dynamics could be different, but authors seem not to have cared for this. They could increase cAMP imaging data analysis.

According to Reviewer’s suggestion we have quantified also ISO responses and added respective graphs to all FRET figures (Fig 3D,H, Fig 5 C,F and Fig 7E). Indeed, in some cases, especially when PDE2 was knocked out in adult myocytes, we could see a significant increase in ISO response- please see Fig. 5C. In Fig 7E where we had single or multiple PDE2A isoforms KOs, we could see a trend for increased ISO response in PDE2A2+A3 KO cells distite overall relatively variable ISO responses (p value was 0.47 compared to Cas9 control when using simple t-test. When doing mixed ANOVA and multiple comparison, this value was above 0.5).

3- the same concept applies to IBMX responses, which are expected to reflect downregulation of specific PDE isoforms - and would therefore be different in cells already devoid of either PDE2 or PDE3

Indeed, IBMX responses were different in some cases as well, especially in the case of PDE2A KO. In our experiments, IBMX responses reflect the contribution of other PDEs (minus PDE2A or PDE3A isoforms which were knock out) and therefore do not reflect the activity of all PDEs. IBMX was used at the end of each experiment to max out the response after ISO and PDE2/3 inhibitor. Therefore, IBMX responses can be deducted as “100 minus ISO response minus PDE2/3 inhibitor response”. Here are the respective IBMX data which we decided not to include into the main manuscript version:

4- can the authors comment as to why PDE2 or PDE3 downregulation does not impact on basal cAMP concentration? 

We thank the Reviewer for this question. Unfortunately, using our FRET protocols in this study we cannot detect basal cAMP concentrations with high accuracy. In neonatal cells, they can be even at the boarder of the sensor detection, whereas in adult cells they have been reported within the sensor detection range (around 1µM basal cytosolic cAMP detected by this sensor, see e.g. Börner, Nikolaev et al. Nat. Protoc. 2011) but exact basal cAMP measurements would require other protocols than used in our study (inhibition of basal cyclase activity, and stimulation with cell permeable cAMP analogue).

Reviewer 2 Report

The English language is overall fine. I have not proof-read the manuscript for tiny errors/typos, I will do that if/when I will receive the revised version.

Author Response

We thank the Reviewer a lot for his/her very positive evaluation of our manuscript. We did our best to address all concerns as detailed point-by-point below.

Fig. 2:

⁃ For the sake of completeness and easiness of reading (given the alignment of the sample legends) I would add NT also to the plots.

We thank the Reviewer for this important suggestion. We have now NT data points to all blot quantifications. Please, see the revised version of the Fig. 2.

⁃ Please add a sentence discussing why in 2B MOI 0.5 has a much fainter band than MOI10, which from the plot should be the MOI associated with the lowest % of expression

Added to Fig 2 legend as suggested: “PDE3A signal in the MOI 0.5 sample appears fainter than for other samples because of a lower protein loading, as evidenced by the GAPDH signal”.

⁃ What does % Cas9 it mean the Y-axis legend? Is it not relative to the housekeeping  GAPDH?

Thank you for this remark. Indeed, it is quantified relative to GAPDH and then normalized to Cas9 transduced group. We have clarified this issue and added new sentences to Fig. 2 and Fig. 3 legends as follows: “PDE expression data were normalized to GAPDH and shown as % of Cas9 samples.”

Fig. 3: please expand the description. In particular, the following points are not clear (at

least to me) :

⁃ At what time points are the compounds administered?

⁃ Are these chosen time point standard? Why do they differ between the samples (e.g., Cilo is shown at approx. 400, 300 and 600 sec in E-G, rest. Should they not be given all at the same time?

We thank the Reviewer for raising this point. We have now added the following explanation to Fig. 3 legend: “Time points of drug applications are indicated by arrows. Drugs were applied after reaching stable baseline for each treatment. Representative FRET traces and data analysis for ISO, BAY and Cilo effects (expressed as % of maximal response).”

⁃ Outside of the effect of IBMX, what are the hypotheses? A drop upon ISO administration in the non-control? A drop upon Cilo/BAY administration only in the controls? That PDE KO impede the increase of cAMP concentration? Why Cas9 alone 3B,3F seems to induce a reduction in the ratio? Please expand a little to make the interpretation easier for the readers less familiar with the topic.

We have now added the following explanation at the bottom of page 5 (lines 189-195): “Transduction with Cas9 alone led to somewhat lower BAY and Cilo responses as compared to NT controls, which is theoretically possible due to the adenoviral treatment, however, these differences did not reach statistical significance (Figure 3D,H).”

Otherwise, we expect that PDE KO impedes the increase of cAMP under the application of the respective PDE inhibitor and in some cases, increases ISO response, e.g due to PDE2 KO – please see also our response to question 2 by the Reviewer 1 and the new panels showing quantification of ISO responses (Fig 3D,H, Fig 5 C,F and Fig 7E). We added a respective phrase on page 8 (lines 230-232): “Interestingly, PDE2A knock-out led also to increased ISO response, suggesting tight regulation of beta-adrenergic receptor/cAMP responses by this PDE (Figure 5C). ”

Fig. 4: For completeness, the barplot should be representative of the blot image, i.e., by having dodged bar for NT/C/KO at each of the 3 times points. The current barplot is hard to related to the blot image.

⁃ What is “Control”? NT at which day? (Which of course does not change, but in the current format it’s confusing)

⁃ Please add p-values for all significant comparisons. I doubt that control vs 4 days is non-significant.

Page 8, 209-213 and Fig. 5:

We thank the Reviewer for this great suggestion. We have now NT values added to bar graphs for each day and calculated statistics to obtain p-values for each day which are all now provide in the revised version of the Fig. 4.

⁃ In the text it says “Functionally, this correlated with completely abolished responses to the PDE2 but not PDE3 inhibitor in PDE2A knock-out cells (Figure 5A-C, Figure S6)” and “…abrogated Cilostamide but not BAY 60-7550 response in PDE3A knockout cardiomyocytes (Figure 5D-F)”. However in Fig. 5 is seems that only the gene specific inhibitor is shown (e.g., BAY for PDE2). So where in the plots (e.g., 5D-F) one can deduce that “…abrogated Cilostamide but not BAY 60-7550 response in PDE3A knock-out cardiomyocytes ”?

These control data are included in Figure S6 (absence of PDE3 inhibitor response reduction in PDE2A KO and vice versa). To clarify this point (which was missing in the second half of this sentence), we have now added: “…but not BAY 60-7550 response in PDE3A knock-out cardiomyocytes (Figure 5D-F, Figure S6).” (please see lines 227-230) 

Fig. 6: please comment on why the bands have no smear for the neonatal while a lot for the adult. Also, what are the multiple bands for e.g., PDE3A1_C,D, which are instead not present in the neonatal?

We have made a comment on this issue in the Fig. 6 legend as kindly suggested by the Reviewer: “To detect all isoforms even at low expression, we performed 14 cycles touch down followed by 35 cycles of regular PCR for both neonatal and adult cells. This may explain high number of potentially unspecific bands appearing next to the most prominent specific band (see Table S2 for expected size) especially in the case of adult cardiomyocytes where the expression of RNAs prone to unspecific amplification might be higher”.

Fig. 7: I am not very familiar with FRET traces and YFP/ CFP sensor, however it looks to me that the curves in B,C and D are very similar (mayor drops upon ISO a n d IBMX, little-to-no change with BAY). How come in the bar plot the double exon KO (which corresponds to 7D, right?) is very different?

Thank you for this question. The difference the figure panels A, B, C, D are ment to show is the magnitude of the BAY response after ISO which is (although relatively small 8-10% of max response) visiable in A, B and C traces and is virtually absent in D traces where both PDE2A isoforms 2A2 and2A3 were knockout out.

Other suggestions:

⁃ L. 274: “specific genomic rearrangements in the targeted locus “ in this context sounds of biological relevance, however it is just due to CRISPR-Cas editing.

Changed to “expected CRISPR-induced genomic rearrangements”  as suggested.

⁃ Expand a bit the discussion:

1) with a paragraph suggesting future approaches/experiments to expand the results of this study.

We thank the Reviewer for this helpful suggestion. We added a new para on page 11 (lines 317-326): “Our approach can be potentially applied to various, especially primary cell types which can be effectively transduced with adenoviral vectors to study the role of individual isoforms in all possible PDE subfamilies. For example, in the case of cardiomyocytes, apart from PDE2A and PDE3A, the role of numerous PDE4 subfamily isoforms can be ad-dressed. This PDE family comprises at least two dozens of different splice variants which regulate local cAMP functions in multiple subcellular microdomains [1,3]. Not only can cAMP and cGMP dynamics be measured by live cell imaging in KO cells. Our system makes it possible to assess functional responses by multiple approaches such as immunoblotting for downstream kinase substrate phosphorylation, contractility measurements, regulation of arrhythmogenic events, pathological cell growth and cell survival.”

2) by discussing the differences/similarities between neonatal vs adult cells. What are the expectations in the 2 types? Why? Are the results confirming them? Which factor can influence the results? etc.

We thank the Reviewer for this helpful suggestion. We added a new half-para on page 11 (lines 313-316) to discuss re differences between neonatal vs adult cells: “While neonatal cells rely only on mitochondrial PDE2A2 isoform, PDE2-dependent regulation of cAMP is more complex and involved all three isoforms localized at different subcellular compartments which have to be more specifically addressed in the future studies.”  

Round 2

Reviewer 2 Report

The authors have satisfactorily addressed my comments. 

I suggest the editors to have a final proof-reading of the manuscript to fine-tune the English language if necessary.